# Concerted and multidisciplinary management of COVID-19 drug therapies during the first two epidemic waves in a tertiary hospital in Marseille, France: Results of the PHARMA-COVID study

Matthieu Peretti[1], Stanislas Rebaudet[2,3], Laurent Chiche[2]*, Hervé Pegliasco[4], Emilie Coquet[1]

1 Service de Pharmacie, Hôpital Européen, Marseille, France, 2 Service d'Infectiologie et de Médecine Interne, Hôpital Européen, Marseille, France, 3 UMR1252 SESSTIM, Aix-Marseille Univ, Inserm, IRD, ISSPAM, Marseille, France, 4 Service de Pneumologie, Hôpital Européen, Marseille, France

* l.chiche@hopital-europeen.fr

**Data Availability Statement:** According to the European Data Protection Regulation (General Data

## Abstract

### Objectives

To evaluate the impact of local therapeutic recommendation updates made by the COVID multidisciplinary consultation meeting (RCP) at the Hôpital Européen Marseille (HEM) through the description of the drug prescriptions for COVID-19 during the first two waves of the epidemic.

### Methods

This retrospective observational study analysed data from the hospital's pharmaceutical file. We included all patients hospitalized for COVID-19 between February 1, 2020 and January 21, 2021 and extracted specific anti-COVID-19 therapies (ST) from computerized patient record, as well as patients' demographic characteristics, comorbidities and outcome. The evolution of ST prescriptions during the study period was described and put into perspective with the updates of local recommendations made during the first (V1, from 2/24/2020 to 7/27/2020), and second (V2, from 7/28/2020 to 1/21/2021) epidemic waves.

### Results

A total of 607 COVID-19 hospitalized patients, 197 during V1 and 410 during V2. Their mean age was 65 years-old, and they presented frequent comorbidities. In total, 93% of hospitalized patients received ST: anticoagulants (90%), glucocorticoids (39%) mainly during V2 (49% vs 17%, P<0.001), and azithromycin (30%) mainly during V1 (71% vs 10%, P<0.001). Lopinavir/ritonavir and hydroxychloroquine were prescribed to 17 and 7 inpatients, respectively, and only during V1. Remdesivir was never administered. A total of 22 inpatients were enrolled into clinical trials.

Protection Regulation, GDPR), data including individual patient characteristics which support the findings of the present study could not be deposited in any integrated repository. However, the full pseudo-anonymised database is available upon reasonable request to Miss Cecile Berthelier (Clinical Research Department) at c. berthelier@hopital-europeen.fr and Miss Margaux Garreau (Ethic Committee) at m.garreau@hopital-europeen.fr.

**Funding:** The author(s) received no specific funding for this work.

**Competing interests:** The authors have declared that no competing interests exist.

## Conclusions

The effective dissemination of evidence-based and concerted recommendations seems to have allowed an optimized management of COVID-19 drug therapies in the context of this emerging infection with rapidly evolving therapeutic questions.

## Introduction

In 2020, the SARS-CoV-2 pandemic was a unique health event that caused the health care system to be overwhelmed in several French regions [1]. The initial "shock" was followed by a rapid reorganization of the management of patients suffering from a new pathology, COVID-19, for which the medical teams had very little data, particularly in terms of medication. The Hôpital Européen Marseille (HEM) is a 610-bed private non-profit hospital located in the heart of Marseille's impoverished districts and having, in particular, emergency, infectious diseases, pneumology and intensive care wards (ICU). From February 2020 onwards, HEM was committed by the regional public health authorities in the response to the COVID-19 pandemic, and played a major role in the care of COVID-19 patients in the Marseille area. At the peak of the first epidemic wave (April 1, 2020), HEM was taking care of up to 75 hospitalized patients, thanks to the involvement of a large part of the establishment's medical community.

As early as March 16, a COVID multidisciplinary consultation meeting (RCP) was set up, bringing together infectiologists, respirologists, internists, ICU physicians, cardiologists, radiologists, biologists, immunologists, epidemiologists and pharmacists. RCP objectives were: (1) to ensure a permanent monitoring of the published literature and guidelines concerning the management of COVID patients; (2) to share observations or difficulties from the field (lack of staff, drugs in short supply. . .); (3) to propose a regular update of local management recommendations; and (4) to organize daily medical discussions of specific COVID patients, not only to homogenize and rationalize rapidly evolving practices, but also to reassure clinicians in a context of off-label prescriptions. Digital tools (WhatsApp group, hospital COVID-19 therapeutic guide on smartphone, webinars) facilitated the real-time dissemination of updated recommendations, as well as advice requests from clinicians and feedback. Pharmacists also had a role in informing about the availability of certain off-label treatments and monitoring the stock status of life-saving drugs.

In order to evaluate the impact of local therapeutic recommendation updates made by the COVID RCP in this context of great initial uncertainty, the objective of the present study was to describe the drug prescriptions for COVID-19 at HEM during the first two waves of the epidemic.

## Methods

The main aim of the COVID multidisciplinary consultation meeting (RCP) at Hôpital Européen Marseille (HEM) was to develop evidence-based, rapid, living guidelines intended to support our clinicians in their decisions about management of severe COVID infection. A living review of the peer-reviewed and grey literature (including preprints articles) was conducted at regular intervals (daily to weekly) and discussed by a panel of experts from various area including infectious diseases specialists, experts in public health as well as other front-line clinicians, specializing in immunology, medical microbiology, critical care, pneumology, internal medicine, hepatology, nephrology, neurology, gastroenterology, and pharmacists. The Grading of Recommendations Assessment, Development and Evaluation (GRADE) approach

was used to assess the certainty of evidence and make recommendations [2]. For all recommendations, the expert panellists reached consensus. Guidelines from various medical societies were considered as potential external validation but never awaited for our internal recommendations. The regular reviewing process was followed by a rapid recommendation development checklist made immediately available to our clinicians via a web-based COVID-19 therapeutic guide on smartphone and WhatsApp groups. The RCP fostered inclusion of patients into ongoing cohorts and trials as much and as early as possible: French COVID Cohort (ClinicalTrials.gov Identifier: NCT04262921); HYCOVID (Hydroxychloroquine Versus Placebo in COVID-19 Patients at Risk for Severe Disease–NCT04325893) [3]; FORCE (Avdoralimab an Anti-C5aR Antibody, in Patients With COVID-19 Severe Pneumonia–NCT04371367) [4]; ANACONDA (Anakinra for COVID-19 Respiratory Symptoms–NCT04364009) [5].

In order to evaluate the impact of local therapeutic recommendation updates made by the COVID RCP, we retrospectively included all patients hospitalized for COVID-19 between February 1, 2020 and January 21, 2021. We extracted specific anti-COVID-19 therapies (ST) from computerized patient record, including those administered in clinical trials in which the hospital participated, as well as patients' demographic characteristics, comorbidities (including BMI, modified Charlson index [6]), and outcome (ICU admission, death). The evolution of ST prescriptions during the study period was described and put into perspective with the updates of local recommendations. We performed a comparative analysis between patients who did (ST+) or did not (ST-) receive specific anti-COVID-19 therapies, excluding anticoagulation, and between patients of the first (W1, from 2/24/2020 to 7/27/2020), and second (W2, from 7/28/2020 to 1/21/2021) epidemic waves in France. The Chi-square and the Student's t tests were used for categorical and quantitative variables, respectively, and computed using RStudio v1.2.5033 for Windows.

This retrospective observational study analysed data from the hospital's pharmaceutical file. The study did not involve humans, but only reused routine patient records. Data access complied with French relevant data protection and privacy regulations. The study thus required neither information nor non-opposition of the included individuals and was approved by the institutional and ethical review board of the Hôpital Européen Marseille (n°2022-01-01).

## Results

During the first two waves, HEM teams managed a total of 607 COVID-19 hospitalized patients, including 197 patients during W1 and 410 during W2 (Fig 1). Inpatients characteristics and outcomes are summarized in Table 1. Their mean age was 65 years-old, and they presented frequent comorbidities: hypertension (43%), diabetes (33%) and obesity (28%). Inpatients mean age and comorbidities were similar between W1 and W2 (Table 1).

In total, 93% of hospitalized patients received ST (i.e. at least one specific anti-COVID-19 therapy) (Fig 1): anticoagulants (90%), including low molecular weight heparin (n = 468, 86%), unfractionated heparin (n = 52, 10%), direct oral anticoagulants (n = 45, 8%) and vitamin K antagonists (n = 8, 1.5%); glucocorticoids (39%), mainly during W2 (49% vs 17%, P<0.001); and azithromycin (30%) (Table 1). Lopinavir/ritonavir and hydroxychloroquine were prescribed to 17 and 7 inpatients, respectively, and only during W1. Remdesivir was never administered in HEM (Table 1). Hydroxychloroquine and lopinavir-ritonavir were administered to 7 and 17 patients, respectively, and only in March-April 2020 (Fig 1). Prescription of azithromycin was frequent during W1 (71% of patients), then it markedly dropped during W2 (10%). A total of 22 inpatients were enrolled into clinical trials testing hydroxychloroquine (NCT04325893) [3] or Avdoralimab (NCT04371367) [4]. The monthly evolution of ST prescriptions followed the recommendations of the COVID RCP (Fig 1).

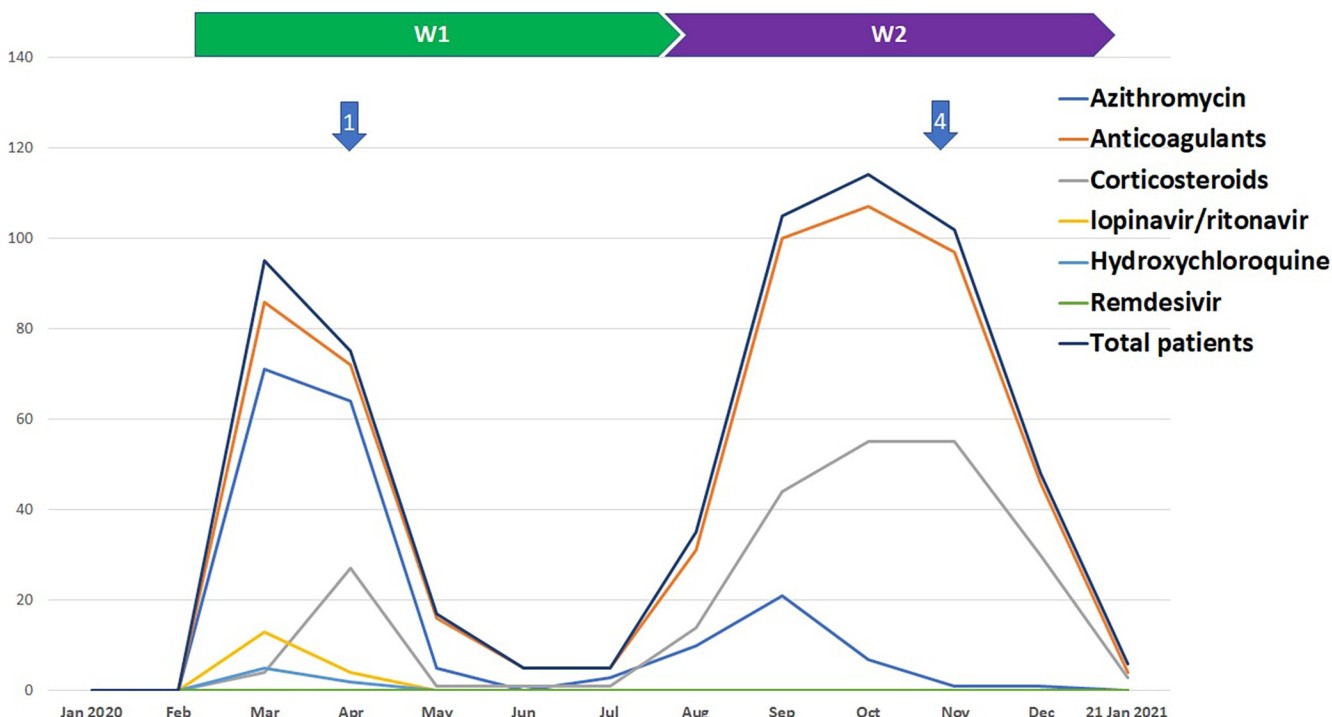

**Fig 1. Monthly trends in prescriptions of specific anti-COVID-19 therapies (excluding clinical trials\*) in hospitalized patients, and main recommendations of the COVID RCP (numbered arrows) during the first two waves (W1 and W2).** Arrow n˚1: HCSP and ISTH [7, 8] official recommendations, and critical analysis of the first publications concerning the lopinavir-ritonavir [9] combination, hydroxychloroquine (HCQ)-azithromycin (AZT) [10] combination and remdesivir [11] by the COVID multidisciplinary consultation meeting (RCP) of the hospital. Arrow n˚2: First randomized trial involving hydroxychloroquine [12] and alerts from pharmacovigilance centres (HCQ ± AZT, lopinavir-ritonavir). Arrow n˚3: first randomized trial of dexamethasone [13]. Arrow n˚4: randomized trial on azithromycin [14]. \*Clinical trials: HYCOVID (NCT04325893, n = 1) [3] and FORCE (NCT04371367, n = 21) [4] evaluating hydroxychloroquine and avdoralimab, respectively.

Patients with comorbidities were more likely to receive a ST (excluding anticoagulation) against COVID-19 (S1 Table): BMI was significantly higher (28.0 vs 26.7, P-value = 0.015), high blood pressure was significantly more frequent (46% vs 37%, P-value = 0.031), and diabetes tended to be more frequent (37% vs 29%, P-value = 0.050) in the ST+ group than in the ST-group.

Non-specific treatments were prescribed to numerous patients: 440 patients (72%) received antipyretics (including paracetamol); 33 (5%) patients received non-steroid anti-inflammatory drugs such as ibuprofen and ketoprofen; 250 patients (41%) received oral antidiabetic treatment or insulin therapy; 360 patients (59%) received anti-hypertensive treatment; and 413 patients (68%) received either antidiabetic or anti-hypertensive treatment.

The overall average length of hospital stay was 13.3 days. A total of 134 inpatients (22%) were admitted to the ICU, and the overall mortality rate was 15% (Table 1). The proportion of patients admitted to the ICU and the mortality rate were similar during W1 and W2 (Table 1). The length of hospital stay was longer in the ST+ group than in the ST- group (16.8 vs 7.9 days, P-value<0.001). Finally, the proportion of patients admitted to the ICU was higher in the ST+ group than in the ST- group (32% vs 6%, P-value < 0.001) (S1 Table).

## Discussion

Hydroxychloroquine, azithromycin and lopinavir-ritonavir were the most used repurposed drugs in different centres around the world for treatment of COVID-19 patients in 2020 [15].

**Table 1. Characteristics and outcome of patients hospitalized with COVID-19, and specific therapies administered during the first (V1) and second (V2) epidemic waves.**

| | Global | First epidemic wave (W1) | Second epidemic wave (W2) | W1 vs W2 (P-value)$ |
|---|---|---|---|---|
| Number of patients hospitalized for COVID-19 | 607 | 197 | 410 | |
| Patients' characteristics (n = 607) | | | | |
| sex ratio (M/F) | 1.7 | 1.9 | 1.6 | 0.417 |
| mean age ± SD (years) | 64.8 ± 16 .2 | 64.2 ± 16.2 | 65.1 ± 16.2 | 0.507 |
| mean BMI ± SD (kg/m2) (n = 529) | 28.1 ± 13.3 | 27.2 ± 5.2 | 28.5 ± 15.6 | 0.342 |
| mean modified Charlson index ± SD | 1.3 ± 2.2 | 1.3 ± 2.3 | 1.3 ± 2.2 | 0.751 |
| obesity (%) (n = 531) | 148 (28%) | 49 (30%) | 99 (27%) | 0.515 |
| diabetes (%) | 203 (33%) | 67 (34%) | 136 (33%) | 0.910 |
| high blood pressure (%) | 259 (43%) | 82 (42%) | 177 (43%) | 0.785 |
| Patients' outcome (n = 607) | | | | |
| ICU admission (%) | 134 (22%) | 48 (24%) | 86 (21%) | 0.402 |
| average length of stay ± SD (days) | 13.3 ± 16 | 13.8 ± 15.5 | 13.1 ± 16.2 | 0.585 |
| mortality rate (%) | 92 (15%) | 31 (16%) | 61 (15%) | 0.877 |
| mean age at death ± SD (years) (n = 92) | 71.5 ± 16.3 | 71.2 ± 15.2 | 71.7 ± 16.8 | 0.903 |
| Specific anti-COVID-19 therapies (ST) (n = 607) | | | | |
| At least 1 ST (including anticoagulant) | 561 (93%) | 182 (92%) | 379 (92%) | 1 |
| At least 1 ST (exclusion of anticoagulant) | 372 (63%) | 154 (78%) | 218 (53%) | <0.001 |
| anticoagulants (%) | 547 (90%) | 174 (88%) | 373 (91%) | 0.379 |
| azithromycin (%) | 183 (30%) | 140 (71%) | 43 (10%) | <0.001 |
| corticosteroids (%) | 235 (39%) | 34 (17%) | 201 (49%) | <0.001 |
| others (%): | 46 (9%) | 23 (12%) | 23 (6%) | 0.001 |
| lopinavir/ritonavir | 17 | 17 | 0 | |
| hydroxychloroquine | 7 | 7 | 0 | |
| remdesivir | 0 | 0 | 0 | |
| tocilizimab | 2 | 2 | 0 | |
| anakinra | 2 | 1 | 1 | |
| convalescent plasma therapy | 1 | 0 | 1 | |
| clinical trials* | 22 | 1 | 21 | |

SD, standard deviation; BMI, body mass index

* HYCOVID (NCT04325893, hydroxychloroquine, n = 1) [3] et FORCE (NCT04371367, avdoralimab, n = 21) [4]

$ Chi-square was used for categorical variables and the Student's t test was used for quantitative variables

In comparison, in our centre, hydroxychloroquine and lopinavir-ritonavir were administered to a very small group of patients and only at the very start of the pandemic, while azithromycin prescription dramatically dropped after the first pandemic wave. Remdesivir was never prescribed to our COVID-19 inpatients during our study period because: this specific drug was not available in our hospital at the beginning the pandemic; it was not judged clinically meaningful according to our implementation process based on the continuous critical reviewing of the available medical literature; and the French drug authorities (*Haute Autorité de Santé*) eventually considered in September 2020 that remdesivir did not bring any improvement in medical benefit in the cure of inpatients with moderate illness, and brought an insufficient medical benefit for those with severe or critical illness [16]. Of note, anti-COVID-19 vaccines and monoclonal antibodies were not available in France during our study period.

The effective dissemination of evidence-based and concerted recommendations during daily multidisciplinary meetings and through digital tools seems to have allowed an optimized

management of COVID-19 drug therapies in the context of this emerging infection with rapidly evolving therapeutic questions. This approach made it possible to initiate exceptional off-label therapies whose interest was then confirmed by clinical trials, and on the contrary to avoid and/or limit uncontrolled prescriptions of therapies secondarily considered ineffective or even deleterious [3, 17], while reassuring hospitalists in the particular context of Marseille, France.

Limitations of the study include the lack of dosage of administered treatments as well as the impossibility to distinguish patients' usual medications against previous comorbidities from those introduced as a consequence of the COVID-19 infection.

The implementation of updated recommendations was eased by immediate diffusion via web application/WhatsApp to front-line clinicians, and daily multidisciplinary discussion in every concerned departments. These new and agile modalities of drug stewardship have been widely used during the following epidemic waves in our hospital, for new anti-COVID therapies such as tocilizumab, convalescent plasma therapy, or monoclonal antibodies. They should, due to their capacity to secure prescriptions, be perpetuated as a new standard of care beyond the pandemic episode.

## Supporting information

**S1 Table. Characteristics and outcomes of patients with (ST+) or without (ST-) anti-COVID-19 specific therapies (excluding anticoagulant) during the first (W1) and second (W2) epidemic waves.**
(DOCX)

## Author Contributions

**Conceptualization:** Matthieu Peretti, Stanislas Rebaudet, Laurent Chiche, Hervé Pegliasco, Emilie Coquet.

**Data curation:** Matthieu Peretti, Stanislas Rebaudet, Laurent Chiche.

**Formal analysis:** Matthieu Peretti, Stanislas Rebaudet.

**Supervision:** Stanislas Rebaudet, Laurent Chiche, Hervé Pegliasco, Emilie Coquet.

**Validation:** Stanislas Rebaudet, Laurent Chiche.

**Visualization:** Matthieu Peretti.

**Writing – original draft:** Matthieu Peretti, Stanislas Rebaudet, Laurent Chiche, Hervé Pegliasco, Emilie Coquet.

**Writing – review & editing:** Matthieu Peretti, Stanislas Rebaudet, Laurent Chiche, Emilie Coquet.

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
