## [Decision Letter · Decision Letter 0]

6 Sep 2022

PONE-D-22-19226Concerted and multidisciplinary management of COVID-19 drug therapies during the first two epidemic waves in a tertiary hospital in Marseille, France: results of the PHARMA-COVID studyPLOS ONE

Dear Dr. chiche,

Thank you for submitting your manuscript to PLOS ONE. Firstly, we would like to apologize for the delay in processing your manuscript. It has been exceptionally difficult to secure reviewers to evaluate your study. We have now received one completed review, which is available below. The reviewer has raised significant scientific concerns about the study that need to be addressed in a revision.

Please note that we have only been able to secure a single reviewer to assess your manuscript. We are issuing a decision on your manuscript at this point to prevent further delays in the evaluation of your manuscript. Please be aware that the editor who handles your revised manuscript might find it necessary to invite additional reviewers to assess this work once the revised manuscript is submitted. However, we will aim to proceed on the basis of this single review if possible.

After careful consideration, we feel that it has merit but does not fully meet PLOS ONE’s publication criteria as it currently stands. Therefore, we invite you to submit a revised version of the manuscript that addresses the points raised during the review process.

We look forward to receiving your revised manuscript.

Kind regards,

Miquel Vall-llosera Camps

Senior Editor

PLOS ONE

Journal Requirements:

3. Please amend your manuscript to include your abstract after the title page.

Reviewers' comments:

Reviewer's Responses to Questions

**Comments to the Author**

1. Is the manuscript technically sound, and do the data support the conclusions?

Reviewer #1: Partly

2. Has the statistical analysis been performed appropriately and rigorously? 

Reviewer #1: Yes

3. Have the authors made all data underlying the findings in their manuscript fully available?

Reviewer #1: No

4. Is the manuscript presented in an intelligible fashion and written in standard English?

Reviewer #1: No

5. Review Comments to the Author

Reviewer #1: Dear Authors,

the Manuscript treated an important topic and the source(s) of data described and used by the authors are interesting, however it has some issues that should be addressed.

1. To be coherent with the objective of the manuscript it might be useful to include in the Introduction or Methods section a paragraph regarding the modality of evidence searching, selection and evaluation. In particular, the authors should describe the methodology, if any, to evaluate the quality or certainty of evidence on medication use in COVID-19 patients in their current clinical practice; if the authors used a standardized methodology please cite it. In additon, how did the authors manage the therapeutic indication coming from the national health authorities regarding the pharmacological treatment of the hospitalized patients affected by COVID-19?

2. The Results section needs to be enriched with more information and better structured, while the Discussion section, in their current form, lacks the appropriate rigor and interpretation to be accepted for publication. Here some suggestions for the authors to improve them:

- the medications included in the ST group should be better described (e.g. what the the authors mean by anticoagulants? Low Molecular Weight Heparins or DOAC? Please better specify this);

- it would be interesting to have more details on medicatons used as non-specific treatments, previously prescribed medications or those co-administered for comorbidities;

- is it possible to know if the patients included in the study were vaccinated or not for COVID-19?

- the use of antipyretics or non-steroid antinflammatory drugs are not mentioned. Have these drugs been used in hospitalized COVID-19 patients?

- the authors should explain the possible reasons because no patients were treated with remdesivir;

- the authors should discuss their results in light of the evidence emerged and evaluated during the considered period;

- please make some comparisons with similar published studies

- have the authors encountered some problems in implementation evidence in the hospital clinical practice? Please describe aspects that should be further improved;

- the authors should describe the limits of the study, also with respect to the possible lack of the above mentioned information.

3. Please check the manuscript for the English language. In Table S1 there is still a sentence in French. Please check the consistency of the terms (e.g. glucocorticoids and corticosteorioids).

Best regards

6. PLOS authors have the option to publish the peer review history of their article (what does this mean?). If published, this will include your full peer review and any attached files.

Reviewer #1: No

---

## [Author Response · Author response to Decision Letter 0]

13 Jan 2023

Response to reviewer

PONE-D-22-19226

Concerted and multidisciplinary management of COVID-19 drug therapies during the first two epidemic waves in a tertiary hospital in Marseille, France: results of the PHARMA-COVID study

Comments to the Author

1. Is the manuscript technically sound, and do the data support the conclusions?

Reviewer #1: Partly

We hope that the modifications following the reviewer’s suggestions will improve significantly the manuscript.

2. Has the statistical analysis been performed appropriately and rigorously?

Reviewer #1: Yes

3. Have the authors made all data underlying the findings in their manuscript fully available?

Reviewer #1: No

With regards to French relevant data protection and privacy regulations, we could not provide individual data, especially in an open-access publication. 

We have added this Data reporting declaration: “According to the European Data Protection Regulation (General Data Protection Regulation, GDPR), data including individual patient characteristics which support the findings of the present study could not be deposited in any integrated repository. However, the full pseudo-anonymised database is available upon reasonable request to the corresponding author.”

4. Is the manuscript presented in an intelligible fashion and written in standard English?

Reviewer #1: No

The manuscript was carefully corrected and edited as suggested.

5. Review Comments to the Author

Reviewer #1: Dear Authors,

the Manuscript treated an important topic and the source(s) of data described and used by the authors are interesting, however it has some issues that should be addressed.

1. To be coherent with the objective of the manuscript it might be useful to include in the Introduction or Methods section a paragraph regarding the modality of evidence searching, selection and evaluation. In particular, the authors should describe the methodology, if any, to evaluate the quality or certainty of evidence on medication use in COVID-19 patients in their current clinical practice; if the authors used a standardized methodology please cite it. In addition, how did the authors manage the therapeutic indication coming from the national health authorities regarding the pharmacological treatment of the hospitalized patients affected by COVID-19?

As suggested, we added a paragraph in the Method section. 

“The main aim of the COVID multidisciplinary consultation meeting (RCP) at Hôpital Européen Marseille (HEM) was to develop evidence-based, rapid, living guidelines intended to support our clinicians in their decisions about management of severe COVID infection. A living review of the peer-reviewed and grey literature (including preprints articles) was conducted at regular intervals (daily to weekly) and discussed by a panel of experts from various area including infectious diseases specialists, experts in public health as well as other front-line clinicians, specializing in immunology, medical microbiology, critical care, pneumology, internal medicine, hepatology, nephrology, neurology, gastroenterology, and pharmacists. The Grading of Recommendations Assessment, Development and Evaluation (GRADE) approach was used to assess the certainty of evidence and make recommendations [3]. For all recommendations, the expert panellists reached consensus. Guidelines from various medical societies were considered as potential external validation but never awaited for our internal recommendations. The regular reviewing process was followed by a rapid recommendation development checklist made immediately available to our clinicians via a web-based COVID-19 therapeutic guide on smartphone and WhatsApp groups. The RCP fostered inclusion of patients into ongoing cohorts and trials as much and as early as possible.” 

[ref] Guyatt GH, Oxman AD, Vist G, Kunz R, Brozek J, Alonso-Coello P, Montori V, Akl EA, Djulbegovic B, Falck-Ytter Y, Norris SL, Williams JW Jr, Atkins D, Meerpohl J, Schünemann HJ. GRADE guidelines: 4. Rating the quality of evidence--study limitations (risk of bias). J Clin Epidemiol. 2011 Apr;64(4):407-15. doi: 10.1016/j.jclinepi.2010.07.017.

2. The Results section needs to be enriched with more information and better structured, while the Discussion section, in their current form, lacks the appropriate rigor and interpretation to be accepted for publication. Here some suggestions for the authors to improve them:

- the medications included in the ST group should be better described (e.g. what the the authors mean by anticoagulants? Low Molecular Weight Heparins or DOAC? Please better specify this);

As suggested, precisions about anticoagulants received by patients were added in the Results section. “anticoagulants (90%), including by declining order low molecular weight heparin (n=468, 86 %), unfractionated heparin (n=52, 10%), direct oral anticoagulants (n=45, 8%) and vitamin K antagonist (n=8, 1.5%).“

- it would be interesting to have more details on medications used as non-specific treatments, previously prescribed medications or those co-administered for comorbidities;

As suggested, precision on the treatments related to comorbidities were added in the Results section: “Non-specific treatments were prescribed to numerous patients: 440 patients (72%) received antipyretics (including paracetamol) ; 33 (5 %) patients received non-steroid anti-inflammatory drugs such as ibuprofen and ketoprofen ; 250 patients (41 %) received oral antidiabetic treatment or insulin therapy ; 360 patients (59 %) received anti-hypertensive treatment ; and 413 patients (68 %) received either antidiabetic or anti-hypertensive treatment.”

We could not differentiate between previously prescribed medications or those administered during the hospital stay for comorbidities. This point was stated as a limitation of the study in the corresponding section of the Discussion. 

- is it possible to know if the patients included in the study were vaccinated or not for COVID-19? 

The period of the study corresponds to the period before access to vaccine in France that was effective since early 2021. The fact that none of our patients was vaccinated was added in the Discussion section. 

- the use of antipyretics or non-steroid antinflammatory drugs are not mentioned. Have these drugs been used in hospitalized COVID-19 patients? 

Yes, these drugs were used. This was added in the results section “Non-specific treatments were prescribed to numerous patients: 440 patients (72%) received antipyretics (including paracetamol) ; 33 (5 %) patients received non-steroid anti-inflammatory drugs such as ibuprofen and ketoprofen ; 250 patients (41 %) received oral antidiabetic treatment or insulin therapy ; 360 patients (59 %) received anti-hypertensive treatment ; and 413 patients (68 %) received either antidiabetic or anti-hypertensive treatment.”

- the authors should explain the possible reasons because no patients were treated with remdesivir;

Remdesivir was never prescribed to our COVID-19 inpatients during our study period because: this specific drug was not available in our hospital at the beginning the pandemic; it was not judged clinically meaningful according to our implementation process based on the continuous critical reviewing of the available medical literature (see Q1); and the French drug authorities (Haute Autorité de Santé) eventually considered in September 2020 that remdesivir did not bring any improvement in medical benefit in the cure of inpatients with moderate illness, and brought an insufficient medical benefit for those with severe or critical illness. This point was added in the Discussion section

- the authors should discuss their results in light of the evidence emerged and evaluated during the considered period;

That is what was intended in the main Figure of the paper showing the evolution of drugs used in parallel of the emergence of significant evidence. 

- please make some comparisons with similar published studies

As suggested, the following comparison was added in the Discussion section. “Hydroxychloroquine, azithromycin and lopinavir-ritonavir were the most used repurposed drugs in different centres around the world for treatment of COVID-19 patients in 2020 [ref]. In comparison, in our centre, hydroxychloroquine and lopinavir-ritonavir were administered to a very small group of patients and only at the very start of the pandemic, while azithromycin prescription dramatically dropped after the first pandemic wave. Remdesivir was never prescribed to our COVID-19 inpatients during our study period” 

[ref] Prats-Uribe A and al. Use of repurposed and adjuvant drugs in hospital patients with covid-19: multinational network cohort study. BMJ. 2021 May.

- have the authors encountered some problems in implementation evidence in the hospital clinical practice? Please describe aspects that should be further improved; 

No, the implementation of updated recommendation was eased by immediate diffusion via web application/WhatsApp to front-line clinicians, and daily multidisciplinary discussion in every concerned departments. This point was added in the Discussion section. 

- the authors should describe the limits of the study, also with respect to the possible lack of the above mentioned information.

The limitation section was completed with the following sentence: “Limitations of the study include the lack of dosage of administered treatments as well as the impossibility to distinguish patients’ usual medications against previous comorbidities from those introduced as a consequence of the COVID-19 infection.”

3. Please check the manuscript for the English language. In Table S1 there is still a sentence in French. Please check the consistency of the terms (e.g. glucocorticoids and corticosteorioids).

Done as suggested.

We thank the reviewer for her/his help in improving our manuscript.

---

## [Decision Letter · Decision Letter 1]

3 Mar 2023

Concerted and multidisciplinary management of COVID-19 drug therapies during the first two epidemic waves in a tertiary hospital in Marseille, France: results of the PHARMA-COVID study

PONE-D-22-19226R1

Dear Dr. chiche,

We’re pleased to inform you that your manuscript has been judged scientifically suitable for publication and will be formally accepted for publication once it meets all outstanding technical requirements.

Kind regards,

Academic Editor

PLOS ONE

Additional Editor Comments (optional):

Reviewers' comments:

Reviewer's Responses to Questions

**Comments to the Author**

1. If the authors have adequately addressed your comments raised in a previous round of review and you feel that this manuscript is now acceptable for publication, you may indicate that here to bypass the “Comments to the Author” section, enter your conflict of interest statement in the “Confidential to Editor” section, and submit your "Accept" recommendation.

Reviewer #1: All comments have been addressed

Reviewer #2: (No Response)

2. Is the manuscript technically sound, and do the data support the conclusions?

Reviewer #1: Yes

Reviewer #2: (No Response)

3. Has the statistical analysis been performed appropriately and rigorously? 

Reviewer #1: Yes

Reviewer #2: (No Response)

4. Have the authors made all data underlying the findings in their manuscript fully available?

Reviewer #1: Yes

Reviewer #2: (No Response)

5. Is the manuscript presented in an intelligible fashion and written in standard English?

Reviewer #1: Yes

Reviewer #2: (No Response)

6. Review Comments to the Author

Reviewer #1: The manuscript was rearranged according to reviewers’ suggestions, thus it can be accepted for publication.

Reviewer #2: (No Response)

7. PLOS authors have the option to publish the peer review history of their article (what does this mean?). If published, this will include your full peer review and any attached files.

Reviewer #1: No

Reviewer #2: No

---

## [Editor Report · Acceptance letter]

8 Mar 2023

PONE-D-22-19226R1 

Concerted and multidisciplinary management of COVID-19 drug therapies during the first two epidemic waves in a tertiary hospital in Marseille, France: results of the PHARMA-COVID study 

Dear Dr. Chiche:

I'm pleased to inform you that your manuscript has been deemed suitable for publication in PLOS ONE. Congratulations! Your manuscript is now with our production department. 

Kind regards, 

on behalf of

Dr. Robert Jeenchen Chen 

Academic Editor

PLOS ONE